# Harnessing Natural Antioxidants for Enhancing Food Shelf Life: Exploring Sources and Applications in the Food Industry

**DOI:** 10.3390/foods12173176

**Published:** 2023-08-23

**Authors:** Carmen Daniela Petcu, Dana Tăpăloagă, Oana Diana Mihai, Raluca-Aniela Gheorghe-Irimia, Carmen Negoiță, Ioana Mădălina Georgescu, Paul Rodian Tăpăloagă, Cristin Borda, Oana Mărgărita Ghimpețeanu

**Affiliations:** 1Faculty of Veterinary Medicine, University of Agronomic Sciences and Veterinary Medicine of Bucharest, 105 Blvd, Splaiul Independentei, 050097 Bucharest, Romania; carmen.petcu@fmvb.usamv.ro (C.D.P.); diana.mihai@fmvb.usamv.ro (O.D.M.); raluca.irimia@fmv.usamv.ro (R.-A.G.-I.); carmen.negoita@fmvb.usamv.ro (C.N.); margarita.ghimpeteanu@fmvb.usamv.ro (O.M.G.); 2Sanitary Veterinary and Food Safety Directorate Bucharest, Ilioara Street No. 16Y, District 3, 032125 Bucharest, Romania; georgescu.madalina-b@ansvsa.ro; 3Faculty of Animal Productions Engineering and Management, University of Agronomic Sciences and Veterinary Medicine of Bucharest, 011464 Bucharest, Romania; paul.tapaloaga@igpa.usamv.ro; 4Faculty of Veterinary Medicine, University of Agricultural Sciences and Veterinary Medicine of Cluj-Napoca, 3-5 Mânăștur St., 400372 Cluj-Napoca, Romania

**Keywords:** natural antioxidants, food industry, food preservation

## Abstract

Consumers are increasingly showing in maintaining a healthy dietary regimen, while food manufacturers are striving to develop products that possess an extended shelf-life to meet the demands of the market. Numerous studies have been conducted to identify natural sources that contribute to the preservation of perishable food derived from animals and plants, thereby prolonging its shelf life. Hence, the present study focuses on the identification of both natural sources of antioxidants and their applications in the development of novel food products, as well as their potential for enhancing product shelf-life. The origins of antioxidants in nature encompass a diverse range of products, including propolis, beebread, and extracts derived through various physical–chemical processes. Currently, there is a growing body of research being conducted to evaluate the effectiveness of natural antioxidants in the processing and preservation of various food products, including meat and meat products, milk and dairy products, bakery products, and bee products. The prioritization of discovering novel sources of natural antioxidants is a crucial concern for the meat, milk, and other food industries. Additionally, the development of effective methods for applying these natural antioxidants is a significant objective in the food industry.

## 1. Introduction

The role of safe food in human nutrition and food preservation concerns was and still is the subject of many research papers. Also, the preferences of consumers who have become more and more attentive to the chemical composition of food products, the preservatives used, the presentation/preservation of food, and their shelf life are more and more present in the research area of food safety [1,2]. Natural antioxidants are substances present in certain foods that help in the protection of the human body against free radicals. Free radicals are unstable molecules that can damage cells and contribute to the occurrence of various conditions, such as premature aging, cancer, and heart disease [3,4,5]. Antioxidants neutralize free radicals with an important role in preventing cell damage. A rich diet in antioxidants can help to reduce the risk of the mentioned diseases and to maintain overall health.

Food products rich in natural antioxidants are (a) vegetables such as spinach, broccoli, cabbage, peppers, carrots, sweet potatoes, tomatoes, beans, lentils, chickpeas, etc.; (b) fruits: forest fruits (blueberries, raspberries, blackberries), cherries, citrus fruits (oranges, lemons, tangerines), pomegranates, kiwi, strawberries, apples; (c) dried fruits and seeds: walnuts, almonds, hazelnuts, chia seeds, flax seeds; (d) cacao and dark chocolate with a high cocoa content; (e) green tea and red tea (rooibos); (f) spices: turmeric, ginger, oregano, parsley, cinnamon; (g) vegetable oils: olive oil, flaxseed oil, sunflower seed oil; and (h) red wine.

A balanced diet rich in fruits and vegetables can provide a wide variety of natural antioxidants. The effects of natural antioxidants are exploited to increase the benefits brought by processed food products. Food additives, including antioxidants, are listed on food labels so that consumers are informed about the composition of the food products [6]. Tocopherols (vitamin E) are widely used to prevent the oxidation of oils and fats. Also, ascorbic acid (vitamin C) is a common antioxidant used in food processing to prevent the discoloration and degradation of oxygen-sensitive nutrients.

The aim of this paper is to systematically examine and evaluate a broad spectrum of natural antioxidants with a specific emphasis on their source and application within the realm of the food industry. Through comprehensive documentation regarding the variety of natural antioxidants utilized in the preservation, quality enhancement, and safety of food, specifically focusing on meat, meat products, dairy, bee products, and plant-based alternative foods, this paper aims to offer significant insights regarding the complex role of natural antioxidants. Additionally, this paper aims to provide an informed perspective on the future trends of antioxidant utilization within the food industry, which will be useful in enhancing the comprehension of current developments and possible innovations in this crucial domain.

## 2. Sources of Natural Antioxidants

More and more preservatives are used in food production to prevent their deterioration via microorganism action. Therefore, the number of antioxidant substances used in the food production process has increased considerably, the result being the extension of the products’ shelf life [1]. A number of factors, such as the presence of oxygen, humidity, high temperature, and light can favor oxidation, and to delay or even prevent food oxidation, oxygen must be removed, and food must be kept in proper storage conditions [7]. Lipids and heme proteins, especially myoglobin, are particularly vulnerable to oxidation. The lipid oxidation process is complex, involving several interacting mechanisms. It is generally accepted that lipid oxidation in animal foods occurs via the autoxidation pathway (initiated by free radicals of often unknown origin), the photooxidation pathway, and the enzymatic pathway (the lipoxygenase pathway). The oxidation of lipids and myoglobin in meat are linked, and both processes are capable of affecting each other. During the oxidation of oxymyoglobin, methemoglobin and hydrogen peroxide are produced, which is necessary to initiate lipid oxidation. The redox stability of myoglobin is altered by aldehyde lipid oxidation products, leading to increased oxidation of oxymioglobin and the consequent development of a covalent adduct with myoglobin [7].

Plant-based extracts can be used successfully as a source of natural antioxidants to prevent the previously mentioned effects and are a good alternative to synthetic preservatives [8,9,10]. Compounds derived from plants are largely phenols or their derivatives. These secondary metabolites have various benefits, including antimicrobial properties [11]. In addition, there are studies that have evaluated the antibacterial effects of aromatic plants (Figure 1) against food-borne bacteria [12].

Thousands of scientific papers on antioxidants used in the food industry have been published over the years. Antioxidants used in the food industry can be classified as natural or synthetic. Natural antioxidants can be used not only in products of animal origin but also in those of vegetal origin. In Table 1, we can observe that antioxidants are found in all foods intended for human consumption of animal or non-animal origin. Most studies have been performed to determine the use of natural antioxidants in the food industry. Natural antioxidants in non-animal origin food are, for example, elder (*Sambucus nigra* L.) in chocolate-covered gingerbread cookies [13], *Arbutus unedo* and *Diospyros kaki* fruits, *Myrtus communis* berry extract, *Acca sellowiana*, or *Crocus sativus* flower for apple juice enrichment [14]. Furthermore, research on antioxidants used in the food industry has begun since 1996, but the largest number of works in this field has appeared in the last 3 years.

The following table shows several vegetal sources from which natural antioxidants can be extracted and used in the food industry (Table 2).

Studies regarding the extension of the shelf-life of food products are a priority for many researchers. Novakovic S. et al., (2019) published results regarding the antioxidant and antimicrobial properties of *Cantharellus cibarius* decoction on some meat products, improving the fresh-keeping function [62]. Also, Indian gooseberry (*Emblica officinalis Gaertn*.) contains major bioactive components, antioxidant and antibacterial, with an important role in the fresh-keeping function on minced chicken meat [63]. 

On the market, there are different antioxidant compounds extracted from natural sources available: EN-FORT^TM^ from rosemary extract, FORTIUM^®^ A from Acerola cherry powder, NaturFORT from rosemary and green tea extracts produced by Kemin Industries (Herentals, Belgium); Boublenza from natural carob produced by SARL Boublenza Agroalimentaire et Produits Agricoles (Tlemcen, Algeria), Herbalox from rosemary extract, Duralox from green tea, acerola, and a variety of other natural ingredients produced by Kalsec (Kalamazoo, MI, USA), and Origanox^TM^ from lemon balm extract produced by RAD Natural Technologies Ltd. (UK) [64,65,66,67].

## 3. Natural Antioxidants Used in Meat and Meat Products

Meat and meat products are recognized as significant contributors to the human diet due to their rich content of proteins, amino acids, lipids, minerals (including zinc, iron, and phosphorus), vitamins, and various other essential nutrients. Consequently, they assume a pivotal function in the field of human nutrition. In spite of the prevalence of modern dietary trends in Western societies promoting decreased or eliminated meat intake, it remains of interest in global meat consumption. In the last two decades, there has been a significant rise of 58% in quantity, resulting in an annual aggregate of 360 million tonnes. The observed upward trend can be primarily attributed to the increase in population, accounting for 54% of the overall impact. The remaining portion of the trend can be ascribed to changes in individuals’ dietary patterns and income levels, leading to a higher per capita consumption of meat [68]. Due to several variables, such as consumer tastes, economics, and geography, the meat industry has undergone significant global transformation. It has been determined that the phenomenon was brought on by a rise in demand for longer meat preservation, a time-tested technique that has made a substantial contribution to the expansion of the global meat industry and is directly associated with product safety and quality. Many methods have been used throughout history to increase the shelf life of meat products, including salting, drying, smoking, fermenting, and canning. Despite the widespread use of both modern and traditional preservation methods, meat rotting continues to be a major problem throughout the entire food supply chain, especially in developing nations. In industrialized countries, losses tend to be concentrated at the market and household levels rather than occurring frequently on farms and in processing plants [68]. Lipid and protein oxidation appears to be the second most important cause of meat deterioration behind microbial spoilage. The presence of oxidative reactions in meat and meat products leads to the development of undesirable flavors, unpleasant odors, changes in color, and deterioration in texture, ultimately leading to a reduction in their shelf life and overall quality. In recent times, there has been a notable increase in research efforts focused on the utilization of antioxidants in the context of meat production with the objective of improving the durability and overall excellence of meat and its by-products. A wide range of natural and synthetic antioxidants have been extensively studied to evaluate their effectiveness in various meat systems [69]. The progress made in the field of antioxidant research has presented meat scientists with the chance to investigate strategies aimed at mitigating the presence of chemical toxins in meat products. The methods discussed in this study encompass a range of strategies, including the implementation of antioxidant interventions, the use of bio-accessibility restriction technology, and the application of moderate thermal processing conditions to reduce the formation of toxins. The latter phenomenon is of particular interest due to the prevailing belief that free radicals, specifically reactive oxygen species (ROS), have a significant impact on the diverse processes that lead to the production of toxins. The application of synthetic antioxidants has been widely utilized in the preservation of meat to counteract negative changes induced by oxidation. Nonetheless, there is a mounting apprehension regarding the potential genotoxic effects of these substances, which has garnered escalated scrutiny. As a result, there has been a notable transition in the present industrial trajectory toward the incorporation of natural antioxidants sourced from various plants. These antioxidants are characterized by their significant polyphenol content, which enables them to efficiently counteract the harmful effects of free radicals [4,70]. 

Numerous research studies have been conducted recently on the use of several natural antioxidant sources to preserve food and enhance the sensory characteristics of meat products. Regarding the antioxidant properties that are useful in food preservation, different fruit extracts exhibited the desired effect. For example, grape pomace seasonings proved to be efficient in cooked and raw beef patties, reducing oxidation. From the three tested grape pomace types, the strongest antioxidant activity was attributed to seedless red wine pomace, all of them being more efficient than sulfite [71]. Acerola fruit powder, rosemary, and licorice extract were also effective in caiman meat nuggets during 120 days of frozen storage, decreasing TBARS (thiobarbituric acid reactive substances) values and improving sensorial characteristics [72]. Cuong et al., (2016), tested the efficiency of annatto in raw pork patties. The research showed a decrease in TBARs, concluding that the extract can be used as a possible antioxidant in meat and meat products [73]. In a similar study, Munekata et al., (2016) tested that peanut skin extract exhibited higher antioxidant properties compared with BHT in raw sheep patties [74]. The use of oil seeds was also efficient in terms of antioxidant properties. In this direction, olive oil was tested in dry fermented sausages. It was concluded that it can inhibit lipid oxidation, additionally improving MUFA (monounsaturated fatty acid) and PUFA (polyunsaturated fatty acid)/SFA (saturated fatty acid) ratios [75]. Sunflower oil was also tested in different meat matrices and design studies. In this regard, Asumin-Bediako et al., (2014) tested the efficiency of the aforementioned oil in UK-style sausages. It was concluded that the fatty acid composition was improved, but the lipid oxidation was not modified [76]. On the other hand, Cardenia et al., (2011) observed an increase in TBAR values and a decrease in the PV (peroxide value) of raw meat when it was used in swine diets [77]. Soybean oil was also tested in Mortadella-type sausages, being observed that the sensory attributes and overall oxidative stability were improved [78]. For example, cinnamon-deodorized aqueous extract improved the stability and color of chicken meatballs with no implication regarding the overall sensory acceptability, being comparable with synthetic antioxidants (BHA, BHT) and ascorbic acid [79]. 

In addition to the inherent antioxidants found in meat, scientific studies propose that the inclusion of dietary compounds rich in antioxidants in animal feeds may enhance their well-being, as well as that of consumers of animal-derived products. Grape pomace (GP), olive cake, and distillers’ grain wastes have been utilized by animal industries in Europe as well as South and Central America due to their widespread availability and suitability for processing. Previous research conducted on live animals has demonstrated that the inclusion of wine and olive oil residues, which are abundant in polyphenols, can enhance the antioxidant capacity, meat quality, and overall well-being of piglets, poultry, and lambs. An additional ecological advantage is the limitation of pollution discharge into terrestrial and aquatic environments [10]. An extensively studied fat-soluble carotenoid with antioxidative properties is α-tocopherol, also known as vitamin E. The main vitamin E constituent in plant leaves is tocopherol, which is found in the thylakoid membranes and chloroplast envelopes close to phospholipids [79]. In this regard, in a study made by Tang et al., (2001), it was observed that α-tocopherol significantly improved the oxidative stability of cooked beef and chicken meat [80]. Grape seed extract, oleoresin rosemary, and water-soluble oregano extract were used for the oxidative and color stability of raw beef and pork patties vacuum-packaged and stored frozen [81].

## 4. Natural Antioxidants Used in the Milk Industry

Dairy product oxidation involves the addition of oxygen atoms or the abstraction of hydrogen atoms from the various compounds present in milk. This process is also associated with the conversion of primary hydroxyl groups to aldehyde and then to carboxylic acid functionality via chemically or biochemically mediated oxidation. In a few circumstances, the oxidation reaction is positive and may lead to an enhancement in product quality, such as the oxidative cross-linking of proteins for gelation and viscosity manipulation. However, in the majority of instances, food oxidation decreases customer acceptance, shortens the product’s shelf life, and in a few instances, may be related to the creation of anti-nutrients and toxicity [82].

Due to their rich range of antioxidant compounds, dairy products are among the most intriguing and promising foods with respect to their potential antioxidant activity. The most significant antioxidants in milk fat are conjugated linoleic acid, β-carotene, vitamins A and E, and coenzyme Q_10_, as shown in Figure 2. Other antioxidant molecules include vitamin D_3_, phospholipids, ether lipids, and maybe 13-methyltetradecanoic acid. Antioxidant properties can also be exhibited by the biopeptides that are produced when cheese is undergoing fermentation or maturation [83,84,85].

On the other hand, it is very important to mention that the endogenous antioxidants of milk can vary depending on different factors. For example, some studies suggest a direct relationship between the milk antioxidant level and the stage of lactation. In this direction, Annie et al. determined how the antioxidant content of milk changes during different phases of lactation in Malabari goats and Vechur cattle [86]. The study concluded that regardless of species, milk produced during the early stage of lactation had much more antioxidants than milk produced during the second and third stages of lactation. Similarly, Kapusta et al. observed that the highest total antioxidant status in milk was on the first days of lactation and decreased gradually on the subsequent days [87].

Another factor that can alter the antioxidant level is the thermal treatment because, during heating, milk components react (e.g., the denaturation and aggregation of whey proteins and formation of new complexes) [83,88].

Moreover, animal nutrition plays an important role in enhancing milk’s antioxidant properties. Modifying the nutritional components of the animal’s diet appears to be a promising strategy for boosting the health-promoting ingredients of animal products [89,90]. 

During the last decade, multiple natural sources have been investigated as possible antioxidant fortifiers for milk products in order to balance out the varying quantities of endogenous antioxidants or to develop new functional foods. Fortifying dairy products with bioactive components (natural antioxidant ingredients) boosts their antioxidant and anti-inflammatory qualities, preventing free radical damage and providing health benefits. The food industry is increasingly interested in phenolic-rich herbs, fruits, vegetables, spices, and other plant materials because they slow lipid oxidation and increase food quality and nutritional value [82]. In this direction, Alenisan et al. examined the impact of red ginseng extract (RGE) on the physicochemical parameters, sensory evaluation, and antioxidant activity of milk, concluding that the antioxidant activity of RGE-supplemented milk samples was greater than that of the control sample [84]. Similar results were obtained by comparing the decoctions of *Matricaria recutita* L. (chamomile) and *Foeniculum vulgare Mill.* (fennel) with potassium sorbate (E202) in yogurts. The samples enriched with natural ingredients had a higher antioxidant activity than those without (and among these, the ones with chamomile decoction). Given the results, the authors concluded that plant decoctions can be used to develop novel yogurts by substituting synthetic preservatives and enhancing the antioxidant qualities of the final product without altering the nutritional profile [9]. Another study in yogurt used grape callus and acidified ethanol extracts from four different grape varieties. On the first day of storage, the yogurts containing grape callus extract demonstrated the highest antioxidant activity compared to all other samples tested. Additionally, gas chromatography examination revealed that the callus yogurt contained at least ten distinct bioactive phenolic components [91]. El-Said et al. tested pomegranate peel extract in stirred yogurt, concluding that an increased percentage can lead to a substantial rise in the antioxidant activity (up to 25 percent). The only associated problem was the viscosity that decreased as the percentage of added extract increased. On the other hand, samples containing 20% and 25% had about the same viscosity [92]. In a similar study, Bertolino et al. added the skins of roasted hazelnuts (*Corylus avellana* L.) from three distinct cultivars to yogurt at two different concentrations (3% and 6%) to boost the dietary fiber and polyphenol content. The 6% hazelnut skin yogurts had the highest functional ability, but the customer preference decreased [93].

Moreover, as illustrated in Table 3, research on natural antioxidants is expanding, targeting new sources. These concern both endogenous and exogenous administration.

## 5. Natural Antioxidants from Bee Products

Bee products have been used for both their nutritional value and therapeutic purposes since ancient times [103]. Bee products, such as honey, pollen, propolis, beeswax, royal jelly, and bee venom, have been some of the most widely used natural products in traditional medicine since ancient times due to their powerful curative qualities and high content of bioactive molecules [103,104].

Plants represent a fundamental source of pollen and nectar in beekeeping because bees harvest them from flowers. Therefore, bee products are closely related to flora sources and, depending on them, acquire different characteristics, including their medicinal properties [105].

Bee products are a potential source of natural antioxidants that can counteract the effects of oxidative stress that underlie the pathogenesis of many diseases. The antioxidant effect of bee products has been widely studied and published [105]. Also, bee products are considered to be a potential source of natural antioxidants, such as flavonoids, phenolic acids, or terpenoids. Nowadays, there is still growing concern for natural substances capable of counteracting the effects of oxidative stress that underlies the pathogenesis of numerous diseases, such as neurodegenerative disorders, cancer, diabetes, and atherosclerosis, as well as the negative effects of various harmful factors and drugs [103,106].

Honey composition and its bioactivity have been reported to mainly depend on the floral source; however, external factors, such as seasonal, geographical, and environmental factors, also play a major role [107,108,109,110].

Several bee products are known for their functional properties, including important antimicrobial and antioxidant actions. Studies were conducted regarding the antioxidant activity (AOA), total polyphenolic content (TPC), and antibacterial action of honey and propolis samples collected from the Greek island of Samothrace, which were applied in vitro either individually or in combination in selected concentrations [108,111].

The supplementation of honey with other bee products, such as propolis, beebread, and pollen, resulted in a significant increase in the flavonoid and phenolic contents and in antiradical activity and reducing power, with the largest effect found for the addition of beebread. Significant linear correlations between the total phenolic and flavonoid contents and antiradical activity and reducing power were found [112]. Some researchers showed that pollen and beebread had stronger antioxidant potential than beeswax and honey [113]. They studied the antioxidant effect of peanut skin extract on honey-roasted peanuts, and it provided high protection against lipid oxidation of the food products [114].

## 6. Natural Antioxidants Used to Obtain Plant-Based Origin Foods

In plant-based food production, natural antioxidants were used in cereal products, edible oils, fruit, vegetables, beverages, etc.

In order to comprehend the oxidative stability of cereal-based products, it is essential to understand the lipid content of the raw materials and the associated changes prior to their incorporation into the products. For example, linoleic acid is a key fatty acid found in nearly all cereal grains. In wheat, corn, and barley, 50–60% of the total lipids are linoleic acid. In addition, in cereal grains, 10–25% and 12–22% of the total lipid are palmitic and oleic acids, respectively. In this direction, hexanal and pentane are excellent indicators of oxidation in cereal-based goods, as LA comprises the majority of total fatty acids [115].

Regarding the use of natural antioxidants, several plant sources have been used in the bakery industry in order to improve some quality or safety traits. For example, Topka et al. tested the effect of elder (*Sambucus nigra* L.) on gingerbread. In this study, chocolate-covered gingerbread cookies were fortified with elderflower dry extract and juice concentrate. The cookies and additives were evaluated for their total phenolic content, phenolic component profile, antioxidant capacity, and advanced glycation end-product production in both free and bound phenolic fractions. The addition of elderflower dry extract to the chocolate-coated gingerbread cookies increased the quantity of phenolic acids in the bound phenolic fraction by up to 28%. The authors concluded that elder products appear to be excellent additives to gingerbread cookies, giving good sensory quality and functional dietary properties [116]. Another study was made by formulating functional biscuits with date fruit fibers grown in the Qassim region. The addition of crude date fibers showed the lowest total phenolic, scavenging activity, and hardness, improving the overall acceptability [13]. Indiarto et al. obtained similar results by adding encapsulated mangosteen (*Garcinia mangostana* L.) peel extract to chocolate biscuits. The 5% concentration showed the most promising physicochemical, organoleptic, and antioxidant effects [117].

Regarding the beverages industry, Pernice et al. investigated the antioxidant properties of bergamot juice in order to obtain a functionalized fruit juice. After the standard production methods, apricot and apple juices supplemented with bergamot exhibited a considerable increase in their antioxidant qualities and a drop in their ascorbic acid concentration [118]. In another study, Gil et al. tested the effect of apple juice enrichment with *Arbutus unedo* and *Diospyros kaki* fruits, *Myrtus communis* berry extract, *Acca sellowiana*, or *Crocus sativus* flower by-products on bioactive compounds and antioxidant activity. The results indicated an increase in both situations [14]. 

Tocopherols are present in all edible oils in varying levels and ratios of α-, β-, γ-, and δ-tocopherols. These are partially eliminated during deodorization, the final stage in edible oil refining. Typically, nature-identical ingredients or natural tocopherol concentrates are used to replace them. As a substitute for tocopherol concentrates, tocopherol-rich specialty oils may be added to or blended with edible oils, such as oil from grapefruit seeds, which are by-products of grapefruit juice production [119,120,121]. Additionally, for a good antioxidant in oils and fats, thermal stability during processing is one of the most critical characteristics. It has been demonstrated that most natural additives have more antioxidant effects and thermal stability than their synthetic counterparts in a variety of edible oils. In this direction, Abd-ELGhany et al. supplemented sunflower oil with olive waste cake extract in order to improve the oil’s thermostability. The results indicated that 200 ppm of olive waste cake extract could effectively protect the sunflower oil at 180 ± 5 °C [122]. Several natural antioxidants, for example, ascorbic acid, a-tocopherol, and beta-carotene, were previously used in bakery products. These natural antioxidants have been proven effective in increasing the shelf life of these bakery products. Even little research has been performed on their role in bakery products, the antioxidative activity of plant extracts, such as curcumin, garcinia, mint, or vanillins, has been analyzed [123].

## 7. Antibacterial Properties of Natural Antioxidant Sources

Natural sources of antioxidants provide benefits beyond their ability to protect against oxidative damage. They possess a wide range of supplementary characteristics that enhance the preservation and quality of food. In addition to prolonging the shelf-life period, these sources frequently have antibacterial properties, hence augmenting safety measures. In this regard, Kaneria et al. conducted a study to assess the antibacterial activity, total phenol content, flavonoid content, 2,2-diphenyl-1-picrylhydrazyl free radical scavenging activity, and phytochemical analysis of five plant species: *Guazuma ulmifolia* L., *Manilkara zapota* L., *Melia azedarach* L., *Syzygium cumini* L., and *Wrightia tomentosa R. and S*. The leaf and fruit of *W. tomentosa* exhibited the most potent antibacterial activity when compared to the other plants that were tested. Gram-positive bacteria exhibited a higher susceptibility compared to Gram-negative bacteria [124]. The study conducted by J. C. López-Romero et al. examined the impact of seasonal variations on the antioxidant, antiproliferative, and antibacterial properties of extracts derived from the leaves of *L. glaucescens Kunth.* The extracts of L. *glaucescens* exhibited significant antioxidant activity and contained phenolic components, particularly quercitrin and epicatechin, which were found in high concentrations. Additionally, they exhibited a modest level of antiproliferative activity against the examined cell lines, with HeLa cells showing the most susceptibility among them. The extracts obtained from *L. glaucescens* exhibited a mild inhibitory effect against *S. aureus*, indicating potential antibacterial action [125]. Additionally, the antibacterial efficacy of an extract derived from olive mill wastewater against both Gram-positive and Gram-negative yeast was reported by S. Abu-Lafi et al. The aforementioned extract encompasses three phenolic molecules, namely, hydroxytyrosol, tyrosol, and oleuropein, which exhibit robust antioxidant properties [126]. In a paper authored by T. Appiah et al., the authors presented findings on the antibacterial properties of methanol extracts derived from *Trametes gibbosa*, *Trametes elegans*, *Schizophyllum commune*, and *Volvariella volvacea*. The study investigated the effects of these extracts on a range of microorganisms, including Gram-positive bacteria, Gram-negative bacteria, and fungi. The methanol extracts of these mushrooms were found to possess secondary metabolites, including tannins, flavonoids, triterpenoids, glycosides, and alkaloids. The extracts of *T. gibbosa* and *T. elegans* exhibited antibacterial action against the tested species. The growth of both Gram-negative and Gram-positive bacteria was effectively inhibited by all four mushroom extracts [127].

The integration of the antibacterial properties found in natural sources of antioxidants into food formulations presents significant potential for transforming food safety methods and enhancing the benchmarks of contemporary food preservation.

## 8. Future Trends Regarding Antioxidants in Food Industry

According to the given literature, additional research is required to determine the optimal dietary combinations and/or minimal levels of antioxidants in products in order to achieve the maximum stability in a final product. This may involve establishing interactions between dietary components and the absorption of the desired antioxidants. Ultimately, more complex feed formulations and a clearer picture are needed in order to best determine the nutritive impact of the by-products that are usually utilized as animal feeds. 

The integration of antioxidant delivery methods into meat formulations is anticipated to promote food safety, shelf-life extension, and nutritional quality, hence meeting customer preferences for healthier and minimally processed alternatives, as technology breakthroughs continue to progress. The utilization of antioxidants in meat products exhibits substantial promise for changing the market in the foreseeable future [10].

Natural antioxidants provide remarkable preservation functions in food products. The capacity to mitigate oxidative processes and impede microbiological proliferation enhances the longevity of products and preserves their overall quality. These multifunctional chemicals serve as innate protectors, shielding against the detrimental effects of environmental elements, such as color fading, texture degradation, and flavor changes. By implementing strategies to reduce lipid oxidation and protein denaturation, manufacturers can effectively maintain product stability and extend shelf life [10].

As it was mentioned for meat and meat products and in the case of milk and dairy products, a health risk assessment is also necessary before using compounds, either endogenous or exogenous [82]. Another problem is that the research results may be challenging to compare due to the use of antioxidant activity assays. A single objective method is needed, as none of the methods is of reference [83]. On the other hand, as it was stated for meat and meat products, a previous oxidation assessment of the conventional product is required to define the necessity of natural antioxidant application. Along with the oxidation assessment, the antioxidant selection rationale must be applied [128]. Functional food designs can also include dairy products with the addition of natural antioxidants [83].

Bee products are an important source of natural antioxidants and can be used as such or by adding other ingredients, thus obtaining nutritionally enriched and valuable foods due to their antioxidant potential. Honey is produced by bees from nectar or plant secretions and therefore contains various substances with antioxidant potential in addition to the antimicrobial and anti-inflammatory effect recognized by numerous specialized studies.

In the plant-based food industry, natural antioxidant addition imposes mainly the same risks mentioned in the other categories. New frying oils, such as high-oleic oils, will be brought to the market as fried food consumption continuously increases. Their application will require research into both the stabilization of foods fried in these oils, as well as their stability during frying conditions. It is reasonable to predict that the need for antioxidants in frying oils will decrease [119].

Moreover, new raw resources, technology, and products are being developed at such a rapid rate that actual progress may exceed predicted progress, which may fall short of expectations. It can be predicted that in the future, plant-based foods will be safer, offer superior sensory qualities, and have an adequate shelf life. Antioxidants will undoubtedly contribute to these enhancements [119].

## 9. Conclusions

The food sector aims to harness natural antioxidants to extend product shelf life. Current research seeks natural antioxidants that do not affect food taste or price. Numerous studies have shown that plant-based antioxidants reduce lipid oxidation and delay alteration in meat and meat products, improving shelf life. Natural antioxidants also preserve the nutritious content of products, minimizing oxidative stress-related health risks. As researchers improve antioxidant delivery systems and find new sources, meat operations may be able to satisfy consumer demands for safer, healthier, and longer-lasting meat products. Milk, dairy, and non-animal products follow the same plant-based antioxidant principles and objectives as meat and meat products. In the recent decade, natural antioxidants have been used in all food industry sectors, enhancing food safety and consumer choice for healthier products. 

In conclusion, natural antioxidants are a hot topic nowadays with more studies confirming their versatility in food products.

## Figures and Tables

**Figure 1 foods-12-03176-f001:**
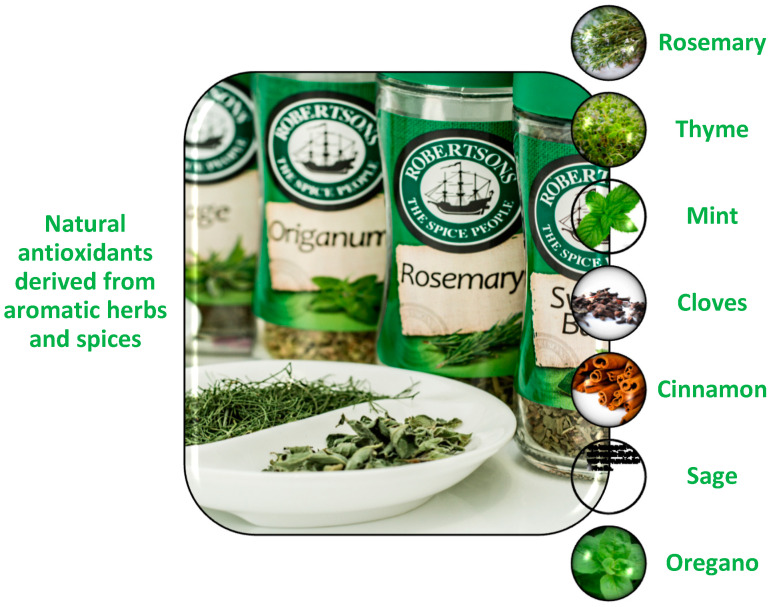
Main categories of natural antioxidants from aromatic herbs and spices. (Illustration made via Microsoft Word/SmartArt (version 2019).)

**Figure 2 foods-12-03176-f002:**
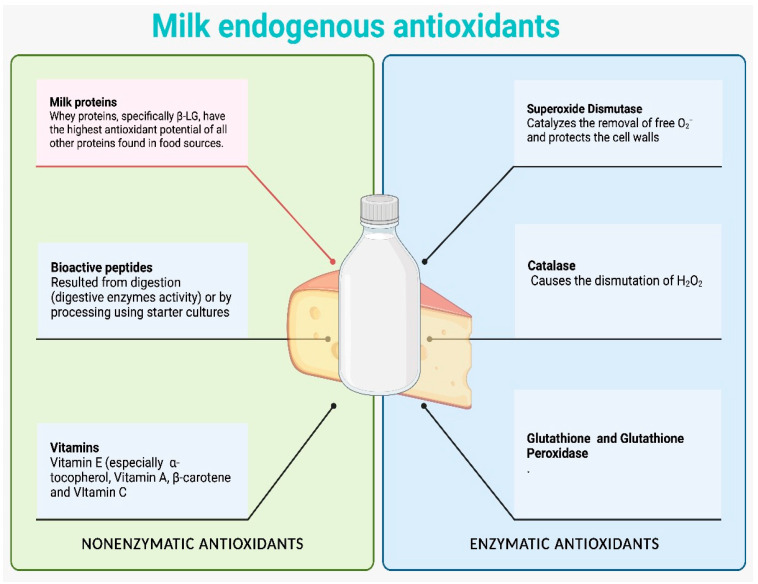
Enzymatic and non-enzymatic antioxidants present in milk (illustration made via www.BioRender.com (accessed on 10 August 2023)).

**Table 1 foods-12-03176-t001:** Bibliometric analysis according to “Web of Science” database, July 2023.

Field of Research	Number of Scientific Publications
Natural antioxidants used in food industry	6852
Natural antioxidants in animal-origin food	218
Natural antioxidants in non-animal-origin food	8
Natural antioxidants in meat	3303
Natural antioxidants in meat products	2018
Natural antioxidants in fishery products	221
Natural antioxidants in milk	1724
Natural antioxidants in dairy products	653
Natural antioxidants used to obtain plant-based foods	41

**Table 2 foods-12-03176-t002:** Vegetal sources used as natural antioxidants in foods.

Source Categories	Plant	Scientific Name	Part Used in Food Industry	References
Aromatic herbs	Basil	*Ocimum basilicum*	Leaves	[15,16]
	Curry	*Murraya koenigii*	Leaves	[1,17]
	Fennel	*Foeniculum vulgare*	Leaves	[9,15]
	Hyssop	*Hyssopus officinalis*	Leaves and secondary branches	[18]
	Lemon balm	*Melissa officinalis*	Leaves	[19]
	Lemon grass	*Cymbopogon citratus*	Leaves	[15]
	Mint	*Mentha spicata*	Leaves	[1,17]
	Myrtle	*Myrtus communis myrtillus*	Leaves	[15,19]
	Oregano	*Origanum vulgare*	Leaves	[1,20]
	Rosemary	*Rosmarinus officinalis*	Leaves and secondary branches	[1,15,18,19]
	Stonecrop	*Sedum sarmentosum Bunge*	Leaves	[21]
	Summer savory	*Satureja hortensis*	Leaves	[1]
	Watercress	*Nasturtium officinale*	Leaves Flower	[15,22]
Algae	Green algae	Green algae	Algae	[15,23]
Dried fruits and seeds	Chestnut	*Castanea*	Leaves Peel	[15,24]
	Peanut	*Arachis hypogaea*	Skin	[1,4]
	Sesame	*Sesamum indicum*	Leaves	[21]
Fruit	Coffee	*Coffea*	Leaves	[15,25,26]
	Date palm	*Phoenix dactylifera*	Pits	[1,27]
	Grape	*Vitis vinifera*	Seed Pomace	[1,15,20,28]
	Kinnow	*Citrus reticulate*	Peel	[1,4]
	Mango	*Mangifera indica*	Peel	[15,29,30]
	Olive	*Olea europaea*	Tree leaves	[1,4,15,31]
	Strawberry	*Fragaria × ananassa*	Leaves Fruit	[4,32]
	Acanthopanax	*Acanthopanax sessiliflorum*	Leaves	[1,21]
	Apple	*Malus pumila*	Peel	[15,33]
	Banana	*Musa*	Peel Fruit	[4,15,34]
	Carob fruit	*Ceratonia siliqua*	Leaves	[35]
	Goji berry	*Lycium barbarum*	Fruit	[15,36]
	Papaya	*Carica papaya*	Seeds	[15,37]
	Peach	*Prunus persica*	Fruit	[15,38]
	Pineapple	*Ananas comosus*	Peel	[15,39]
	Pomegranate	*Punica granatum*	Flower Peel	[4,15,20,40,41,42]
	Blueberry	*Vaccinium sect. Cyanococcus*	Leaves Fruit	[15,43]
	Cape gooseberry	*Physalis peruviana*	Leaves Fruit	[15,44,45]
	Chokeberry	*Aronia melanocarpa*	Leaves Fruits	[46,47,48,49]
	Cranberry	*Vaccinium myrtillus*	Leaves Fruit	[25]
Herbal teas	Ginkgo biloba	*Ginkgo biloba*	Leaves	[15,50]
	Butterbur	*Petasites japonicus Maxim*	Leaves	[1,51]
	Chamomile	*Matricaria recutita* L.	Flower	[9,15]
	Green tea	*Camellia sinensis*	Leaves	[1]
	Nettle	*Urtica dioica*	Leaves Flower	[1,19,52,53,54]
	Roselle	*Hibiscus sabdariffa*	Flower water	[1]
Spice	Fenugreek	*Trigonella foenum-graecum*	Seed	[1]
	Clove	*Eugenia caryophylata*	Bud	[1,20]
	Black seed	*Nigella sativa*	Seeds	[40]
	Cinnamon stick	*Cinnamomum burmannii*	Cortex	[1,20]
	Cinnamon	*Cinnamomum verum*	Bark	[1,15,40]
	Ginger	*Zingiber officinale*	Rhizome Flowering head	[1,55]
Vegetable	Licorice	*Glycyrrhiza glabra*	Root	[40]
	Lotus	*Nelumbo nucifera*	Rhizome knot Leaves	[1,56,57]
	Potato	*Solanum tuberosum*	Peel	[1]
	Pumpkin	*Cucurbita moschata Duch*	Leaves	[1,21,51]
	Bok choy	*Brassica campestris* L. ssp. *chinensis*	Leaves	[1,21,51]
	Broccoli	*Brassica oleracea* L. *var. italica*	Flowering head	[1,21,51,58]
	Carrot	*Daucus carota*	Peel	[15,59]
	Chamnamul	*Pimpinella brachycarpa*	Leaves	[1]
	Chinese chives/Leek	*Allium tuberosum Rottler ex Spreng*	Leaves	[21,51]
	Crown daisy	*Chrysanthemum coronarium*	Leaves	[21,51]
	Eggplant	*Solanum melongena*	Peel	[15,60]
	Eleutherine	*Eleutherine americana*	Bulb	[1,20]
	Fatsia	*Aralia elata Seem*	Leaves	[21,51]
	Garlic	*Allium sativum*	Aerial parts Bulb	[1,40,55]
	Onion	*Allium cepa* L.	Bulb	[55]
	Soybean	*Glycine max* L. *Merr*	Leaves	[4,51]
	Sweet potato	*Ipomoea batatas*	Peel	[15,61]
	Tomato	*Solanum lycopersicum*	Pulp	[15,32]

**Table 3 foods-12-03176-t003:** Literature review on the effects of some natural antioxidant sources on milk by diet supplementation in ruminants and exogenous application in milk products.

Source Categories	Antioxidant/ Plants with Antioxidant Activity	Endo- (EN)/ Exogenous (EX)	Species/Product	Dose/Treatment	Tested Effect	Results	Ref.
FRUITS	Grape pomace (GP)	EN	Bovine (Holstein-Friesian)	Control and 7.5% GP-supplemented diet	Whole-blood transcriptome, milk production, and composition	GP supplementation affected 40 genes in the transcriptome, but milk production and composition were not different between groups.	[94]
		EN	Sheep (dairy ewes)	Control and 10% GP-supplemented diet	Milk yield, chemical-nutritional characteristics, total phenolic compounds, antioxidant activity, fatty acids, and proteins profile	Increase in monounsaturated fatty acids and a decrease in medium-chain saturated fatty acids. It can be fed to lactating ewes without changing milk gross composition but drastically modifying fatty acid profile.	[95]
VEGETABLES	Anthocyanin-rich purple corn (*Zea mays* L.)	EN	Goats (Saanen dairy goats)	Control (sticky corn stover silage) and anthocyanin-rich purple corn diet (TPSS)	Transferring anthocyanin composition to the milk and increasing antioxidant status of lactating dairy goats	Anthocyanin-rich purple corn can increase the antioxidant levels in lactating dairy goats’ milk.	[96]
SPICES, HERBAL TEAS, AND AROMATIC HERBS	Distillate rosemary leaves	EN	Goats (Murciano-Granadina goats)	Basal diet (BD), 10% and 20% (BD) with 50% barley and 50% distilled leaves	Polyphenolic profile of the goats’ milk during the physiological stages of gestation and lactation	Increased polyphenolic content in goat’s milk and kid’s plasma	[97]
	Herbal extract of green mate (*Ilex paraguariensis*), clove (*Syzygium aromaticum*), and lemongrass (*Cymbopogon citratus*)	EX	Fermented milk (FM) with/without sweet potato pulp	N—natural fermented milk, E—phenolic-rich herbal extract (1 g/100 g^−1^), EB—phenolic-rich herbal extract (1 g/100 g^−1^) and sweet potato pulp (15 g/100 g^−1^), and B—sweet potato pulp	Proximate composition, pH, acidity, instrumental texture profile, total phenolic content (TPC), antioxidant activity (AA) of all formulations were measured and sensory attributes	FM with sweet potato pulp showed the best sensory acceptability, while the lyophilized extract containing 87.5% clove and 12.5% green mate improved the antioxidant activity and total phenolic content.	[98]
	Yerba mate (*Ilex paraguariensis A. St.-Hil*.-YM)	EX	Fresh cheese	0.0% (control), 0.5% (FC5), 1.0% (FC10) and 2.0% (FC20)	Bioactive compound concentration, antioxidant activity, color, texture, structure, and sensory acceptance	YM conferred antioxidant activity to FC and affected the color, texture, and structure	[99]
	RGE	EX	Milk (M) and yogurt (Y)	Control and 2% red ginseng extract in milk (RM) and yogurt (RY)	Antioxidant and antigenotoxic effects	Red ginseng can boost dairy products’ antioxidant and antigenotoxic effects.	[100]
	*Rosa spinosissima* fruits extract	EX	Yogurt	Control, 0.1%, and 0.2% extract	Physicochemical properties, microbiology, and antioxidant properties	Significantly affected the yogurts’ antioxidant properties	[101]
	*Solenostemma argel* Hayne leaf extract (ALE)	EX	Yogurt	0.0, 0.1, and 0.2 g/100 mL	Physicochemical, antioxidant, and sensory qualities	ALE increased the antioxidant properties.	[102]

## Data Availability

Data are contained within the article.

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
