# Peer review of "Harnessing Natural Antioxidants for Enhancing Food Shelf Life: Exploring Sources and Applications in the Food Industry"

_foods, 2023, doi:10.3390/foods12173176_

Round 1

Reviewer 1 Report

Abstract: The main objective of this review was to identify, on the one hand, natural sources of antioxidants, and on the other hand, their uses in obtaining food products. I suggest the following as main objective…  to identify, on the one hand, natural sources of antioxidants, and on the other hand, their uses in obtaining new food products and to improve their shelf life.

Please specify in Table 1, which natural antioxidants in non-animal origin food are?

Organize Table 2 and Table 3 according to source categories; example a) vegetables, b) fruits, c) dried fruits and seeds, d) cacao and dark chocolate with a high cocoa content; e) herbal teas, f) spices, g) vegetable oils, h) aromatic herbs, etc. This allows to the reader a better understanding of the natural antioxidants sources.

Figure 2, please correct “cell wal” by “cell wall”, please check all this typing errors through the whole document

Figure 2, if possible, use the MicrosoftWord/SmartArt to improve the figure because it is blurry as it is.

Line 219-220, 235; there are two types of cites, Kapusta et al. …. and [77], according to the general considerations for manuscript preparation, citations In the text, reference numbers should be placed in square brackets [ ], please check this in the whole manuscript.

Minor editing of English language required

Reviewer 2 Report

For a review, I think the paper conforms to the way of writing. But for the topic, I think the title "Natural antioxidants: sources and applications in the food industry" is too big. There are so many natural antioxidants and different sources in nature. If the author focuses on this direction, it will be too big. The theme of the paper submission is "Emerging Natural Antimicrobials for Food Protection and Shelf Life Extension", so the paper is best used in the field of Shelf Life Extension with Natural antioxidants.

Some points:

1 In the introduction section, please explain the reason and scope of this review. You can emphasize the classification of polyphenols used in fresh-keeping applications, or according to their functions, such as anti-oxidation and antibacterial, etc.

Section 2 “Sources of natural antioxidants” should focus on the natural sources of polyphenols used for fresh-keeping function, rather than the plant sources of all polyphenols.

In the third part "Natural antioxidants used in meat and meat products", please focus on the application direction of antioxidants.

It is recommended that subsequent applications be divided by function.

A bacteriostatic application can be added after the sixth part.

In the seventh part, the application in the field of preservation is proposed.

Conclusion Please simplify.

Author Response

Dear Reviewer,

Thank you kindly for your time and attention in reviewing the manuscript. All your comments were very helpful in providing a comprehensive view of the subject. Thus, we present our responses punctually:

Point 1: For a review, I think the paper conforms to the way of writing. But for the topic, I think the title "Natural antioxidants: sources and applications in the food industry" is too big. There are so many natural antioxidants and different sources in nature. If the author focuses on this direction, it will be too big. The theme of the paper submission is "Emerging Natural Antimicrobials for Food Protection and Shelf Life Extension", so the paper is best used in the field of Shelf Life Extension with Natural antioxidants.

Response 1: We have changed the title according to your suggestions (page 1): Harnessing Natural Antioxidants for Enhancing Food Shelf Life: Exploring Sources and Applications in the Food Industry.

Some points:

Point 2: 1 In the introduction section, please explain the reason and scope of this review. You can emphasize the classification of polyphenols used in fresh-keeping applications, or according to their functions, such as anti-oxidation and antibacterial, etc.

Response 2: We have added the aim of the review according to your suggestions, as follows (page 2):

“The aim of this paper is to systematically examine and evaluate a broad spectrum of natural antioxidants, with a specific emphasis on their source and application within the realm of the food industry. Through a comprehensively documentation regarding the variety of natural antioxidants utilized in the preservation, quality enhancement, and safety of food, specifically focusing on meat, meat products, dairy, bee products, and plant-based alternative foods, the paper aims to offer significant insights regarding the complex role of natural antioxidants. Additionally, the paper aims to provide an informed perspective on the future trends of antioxidant utilization within the food industry, which will be useful in enhancing comprehension of current developments and possible innovations in this crucial domain.”

Point 3: Section 2 “Sources of natural antioxidants” should focus on the natural sources of polyphenols used for fresh-keeping function, rather than the plant sources of all polyphenols.

Response 3: We included at page 9 the following:

Studies regarding the extension of the shelf-life of food products are a priority for many researchers. Novakovic S. et al. (2019) published result regarding antioxidant and antimicrobial properties of Cantharellus cibarius decoction on some meat products improving the fresh-keeping function [62]. Also, Indian gooseberry (Emblica officinalis Gaertn.) contains major bioactive components, antioxidant and antibacterial with an important role for fresh-keeping function on minced chicken meat [63].

Point 4: In the third part "Natural antioxidants used in meat and meat products", please focus on the application direction of antioxidants. It is recommended that subsequent applications be divided by function.

Response 4: We have modified according to your suggestion in pages 9-11, as follows:

“Numerous researches have been conducted recently on the use of several natural antioxidant sources to preserve food and enhance the sensory characteristics of meat prod-ucts. Regarding the antioxidant properties which are useful in food preservation, different fruit extracts exhibited the desired effect. For example, grape pomace seasonings proved to be efficient in cooked and raw beef patties, reducing oxidation. From the three tested grape pomaces types, the strongest antioxidant activity was attributed to seedless red wine pomace, all of them being more efficient than sulfite[63]. Acerola fruit powder, rosemary and licorice extract were also effective in caiman meat nuggets during 120 days of frozen storage, decreasing TBARS (thiobarbituric acid reactive substances) values and improving the sensorial characteristics [64]. Cuong et al. (2016), tested the efficiency of annatto in raw pork patties. The research showed a decrease of TBARs, concluding that the extract can be used as a possible antioxidant in meat and meat products[65]. In a similar study, Munekata et al. (2016) tested the peanut skin extract exhibited higher antioxidant proper-ties compared with BHT in raw sheep patties[66]. The use of oil seeds was also efficient in terms of antioxidant properties. In this direction, olive oil was tested in dry fermented sausages. It was concluded that it can inhibit lipid oxidation, additionally improving the MUFA (Monounsaturated Fatty Acid) and PUFA (Polyunsaturated Fatty Acid)/SFA (Saturated Fatty Acid) ratios[67]. Sunflower oil was also tested in different meat matrices and design studies. In this regard, Asumin-Bediako et al. (2014) tested the efficiency of the aforementioned oil in UK-style sausages. It was concluded that the fatty acid composition was improved but the lipid oxidation was not modified[68]. On the other hand, Cardenia et al. (2011) observed an increase in TBARs values and a decrease in PV (Peroxide Value) of raw meat when it was used in swine diets[69]. Soybean oil was also tested in Mortadel-la-type sausages, being observed that the sensory attributes and overall oxidative stability were improved[70]. For example, cinnamon deodorised aqueous extract improved the stability and colour of chicken meatballs with no implication regarding the overall sensory acceptability, being comparable with synthetic antioxidants (BHA, BHT) and ascorbic acid [71].

In addition to the inherent antioxidants found in meat, scientific studies propose that the inclusion of dietary compounds rich in antioxidants in animals' feed may enhance their well-being, as well as that of consumers of animal-derived products. Grape pomace (GP), olive cake, and distillers' grain wastes have been utilized by animal industries in Europe as well as South and Central America due to their widespread availability and suitability for processing. Previous research conducted on live animals has demonstrated that the inclusion of wine and olive oil residues, which are abundant in polyphenols, can enhance the antioxidant capacity, meat quality, and overall well-being of piglets, poultry, and lambs. An additional ecological advantage is the limitation of pollution discharge into terrestrial and aquatic environments[10]. An extensively studied fat-soluble carotenoid with antioxidative properties is α-tocopherol, also known as vitamin E. The main vitamin E constituent in plant leaves is tocopherol, which is found in the thylakoid mem-branes and chloroplast envelopes close to phospholipids[71]. In this regard, in a study made by Tang et al. (2001), in was observed that α-tocopherol significantly improved the oxidative stability of cooked beef and chicken meat[72].”

Point 5: A bacteriostatic application can be added after the sixth part.

Response 5: We have added the section (page 17), as you suggested:

“7. Antibacterial properties of natural antioxidant sources

Natural sources of antioxidants provide benefits beyond their ability to protect against oxidative damage. They possess a wide range of supplementary characteristics that enhance the preservation and quality of food. In addition to prolonging the shelf-life period, these sources frequently have antibacterial properties, hence augmenting safety measures. In this regard, Kaneria et al conducted a study to assess the antibacterial activity, total phenol content, flavonoid content, 2,2-diphenyl-1-picrylhydrazyl free radical scavenging activity, and phytochemical analysis of five plant species: Guazuma ulmifolia L., Manilkara zapota L., Melia azedarach L., Syzygium cumini L., and Wrightia tomentosa R.& S. The leaf and fruit of W. tomentosa exhibited the most potent antibacterial activity when compared to other plants that were tested. Gram-positive bacteria exhibited a higher susceptibility compared to Gram-negative bacteria [124]. The study conducted by J. C. López-Romero et al. examined the impact of seasonal variations on the antioxidant, antiproliferative, and antibacterial properties of extracts derived from the leaves of L. glaucescens Kunth. The extracts of L. glau-cescens exhibited significant antioxidant activity and contained phenolic components, par-ticularly quercitrin and epicatechin, which were found in high concentrations. Additional-ly, they exhibited a modest level of antiproliferative activity against the examined cell lines, with HeLa cells showing the most susceptibility among them. The extracts obtained from L. glaucescens exhibited a mild inhibitory effect against S. aureus, indicating potential antibac-terial action [125]. Additionally, the antibacterial efficacy of an extract derived from olive mill wastewater against both Gram-positive and Gram-negative yeast was reported by S. Abu-Lafi et al. The aforementioned extract encompasses three phenolic molecules, namely hydroxytyrosol, tyrosol, and oleuropein, which exhibit robust antioxidant properties [126]. In a paper authored by T. Appiah et al., the authors presented findings on the antibacterial properties of methanol extracts derived from Trametes gibbosa, Trametes elegans, Schizophyl-lum commune, and Volvariella volvacea. The study investigated the effects of these extracts on a range of microorganisms, including Gram-positive bacteria, Gram-negative bacteria, and fungi. The methanol extracts of these mushrooms were found to possess secondary metab-olites, including tannins, flavonoids, triterpenoids, glycosides, and alkaloids. The extracts of T. gibbosa and T. elegans exhibited antibacterial action against the tested species.  The growth of both Gram-negative and Gram-positive bacteria was effectively inhibited by all four mushroom extracts [127].

The integration of the antibacterial properties found in natural sources of antioxi-dants into food formulations presents significant potential for transforming food safety methods and enhancing the benchmarks of contemporary food preservation.”

Point 6: In the seventh part, the application in the field of preservation is proposed.

Response 6: We have added and modified at page 18, as follows:

The integration of antioxidant delivery methods into meat formulations is anticipated to promote food safety, shelf-life extension and nutritional quality, hence meeting customer preferences for healthier and minimally processed alternatives, as technology breakthroughs continue to progress. The utilization of antioxidants in meat products exhibits substantial promise for changing the market in the foreseeable future [10].

Natural antioxidants provide remarkable preservation functions in food products. The capacity to mitigate oxidative processes and impede microbiological proliferation enhances the longevity of products and preserves their overall quality. These multifunctional chemicals serve as innate protectors, shielding against the detrimental effects of environmental elements such as color fading, texture degradation, and flavor changes. By implementing strategies to reduce lipid oxidation and protein denaturation, manufacturers can effectively maintain product stability and extend its shelf life [10].

Point 7: Conclusion Please simplify.

Response 7: We have simplified the conclusion (page 19) as follows:

“The food sector aims to harness natural antioxidants to extend product shelf life. Current research seeks natural antioxidants that do not affect food taste or price. Numerous studies have shown that plant-based antioxidants reduce lipid oxidation and delay alteration in meat and meat products, improving shelf life. Natural antioxidants also preserve the nutritious content of products, minimizing oxidative stress-related health risks. As researchers improve antioxidant delivery systems and find new sources, meat operations may be able to satisfy consumer demands for safer, healthier, and longer-lasting meat products. Milk, dairy, and non-animal products follow the same plant-based anti-oxidant principles and objectives as meat and meat products. In the recent decade, natural antioxidants have been used in all food industry sectors, enhancing food safety and consumer choice for healthier products.

In conclusion, natural antioxidants are a hot topic nowadays, with more studies confirming their versatility in food products.”

Kind regards,

The authors

Reviewer 3 Report

There are many points that need to clear 

1-The aim  of the work must be cleared

2-green algae do not contain leaves

3-in Table 2 must be added commercial products

4- add some examples of antioxidant compounds extracted from natural sources

5- Authors mentioned in aims ( The identification of  new sources of natural antioxidants is a priority for the food industry   What industries use natural products and their products?

6- Authors mentioned in aims ( and the method of processing and preservation of natural antioxidants represents an important objective for food safety   what are the methods used for antioxidants compounds used in food preservation?  

7- what meat, bee and plant products used natural antioxidants?  

8- it must be title divided into subtitle   

9- the conclusion is good 10- references appropriate

Author Response

Dear Reviewer,

Thank you kindly for your time and attention in reviewing the manuscript. All your comments were very helpful in providing a comprehensive view of the subject. Thus, we present our responses punctually:

Point 1: The aim of the work must be cleared

Response 1: We have added the aim of the review according to your suggestions (page 2), as follows:

 “The aim of this paper is to systematically examine and evaluate a broad spectrum of natural antioxidants, with a specific emphasis on their source and application within the realm of the food industry. Through a comprehensively documentation regarding the variety of natural antioxidants utilized in the preservation, quality enhancement, and safety of food, specifically focusing on meat, meat products, dairy, bee products, and plant-based alternative foods, the paper aims to offer significant insights regarding the complex role of natural antioxidants. Additionally, the paper aims to provide an informed perspective on the future trends of antioxidant utilization within the food industry, which will be useful in enhancing comprehension of current developments and possible innovations in this crucial domain.

Point 2: Green algae do not contain leaves,

Response 2: We filled in the table 2 the term algae at page 4.

Point 3: In Table 2 must be added commercial products. Add some examples of antioxidant compounds extracted from natural sources:

Response 3: We included at page 9 the following:

There are available on the market different antioxidant compounds extracted from natural sources. EN-FORTTM from rosemary extract, FORTIUM® A, from Acerola cherry powder , NaturFORT from rosemary and green tea extracts, produced by Kemin Industries; Boublenza, from natural carob, produced by SARL Boublenza Agroalimentaire et Produits Agricoles, Herbalox from rosemary extract, Duralox from green tea, acerola and a variety of other natural ingredients, produced by Kalsec, OriganoxTM  from lemon balm extract produced by RAD Natural Technologies Ltd. [64-67].

Point 4: Authors mentioned in aims (The identification of new sources of natural antioxidants is a priority for the food industry What industries use natural products and their products? Authors mentioned in aims (and the method of processing and preservation of natural antioxidants represents an important objective for food safety   what are the methods used for antioxidants compounds used in food preservation?  

Response 4: We have made the changes at page 1, as suggested: “Consumers are increasingly expressing a heightened interest in maintaining a healthy dietary regimen, while food manufacturers are striving to develop products that possess an extended shelf-life to meet the demands of the market. Numerous studies have been conducted to identify natural sources that contribute to the preservation of perishable food derived from animals and plants, thereby prolonging its shelf life. Hence, the present study focuses on the identification of both natural sources of antioxidants and their applications in the development of novel food products, as well as their potential for enhancing product shelf-life. The origins of antioxidants in nature encompass a diverse range of products, including propolis, beebread, and extracts derived through various physical-chemical processes. Currently, there is a growing body of research being conducted to evaluate the effectiveness of natural antioxidants in the processing and preservation of various food products, including meat and meat products, milk and dairy products, bakery products, and bee products. The prioritization of discovering novel sources of natural antioxidants is a crucial concern for the meat, milk and other food industries. Additionally, the development of effective methods for applying these natural antioxidants is a significant objective in the food industry.”

Point 5: what meat, bee and plant products used natural antioxidants?  

Response 5:

We added at page 11 the following:

Grape seed extract, oleoresin rosemary and water-soluble oregano extract were used for the oxidative and color stability of raw beef and pork patties vacuum packaged and stored frozen [ 81].

We added at page 16 the following:

It was studied the antioxidant effect of peanut skins extract on honey roasted peanuts and it was provided high protection against lipid oxidation of the food products.[114]

We added at page 17 the following:

Several natural antioxidants, for example ascorbic acid, a-tocopherol, and beta-carotene were previously used in bakery products. These natural antioxidants have been proven effective in increasing the shelf life of these bakery products. Even little research has been done on their role in bakery products, the antioxidative activity of plant extracts such as curcumin, garcinia, mint  or vanillins has been analysed [123].

Point 6: It must be title divided into subtitle   

Response 6: We have changed the title according to your suggestions (page 1): Harnessing Natural Antioxidants for Enhancing Food Shelf Life: Exploring Sources and Applications in the Food Industry.

Kind regards,

The authors

Reviewer 4 Report

Dear Authors

Thank you for your fine work, please notice that 

1.               What is the main question addressed by the research?

The manuscript mainly conducted to test the efficiency of natural antioxidants in processing and preservation of  Food industry such as : meat and  meat products, milk and dairy products, bakery products and bee products

2.     Do you consider the topic original or relevant in the field? Does it address a specific gap in the field? Almost yes.

Seems the respected authors can improve the paper for more scientific sounds , by adding more discussion on Food chemistry on this subject.

3.     What does it add to the subject area compared with other published material?

4.     Not very specific review article, only  can evaluate the manuscript as the new paper in this regard, which probably claimed on the number of recent studies, while they can improve it by adding more food industries included to the benefit of natural antioxidants, for  example mentioning the fact that  natural antioxidants have also been applied to enhance the stability of edible oils, ……

What specific improvements should the authors consider regarding the methodology? What further controls should be considered? The methodology is acceptable in this fine form!

Are the conclusions consistent with the evidence and arguments presented and do they address the main question posed?

6. Are the references appropriate? Highly recommend referring more updates references in recent years.  The paper can be improving the sound of article by adding more relevant papers published in recent years.

7. Please include any additional comments on the tables and figures.

The quality of Figure 2 should be improved, and  in table 1, the date of survey on the  mentioned numbers of published papers is obviously needed!

Round 2

Reviewer 3 Report

Accept in present form